



# Brief communication: Testing a portable Bullard-type temperature lance confirms highly spatially heterogeneous sediment temperatures under shallow water bodies in the Arctic

Frederieke Miesner[1], William Lambert Cable[1], Pier Paul Overduin[1], and Julia Boike[1,2]

[1]Alfred Wegener Institure Hlemholtz Centre for Marine and Polar Research, Permafrost Research, Potsdam, Germany
[2]Humboldt University Berlin, Department of Geosciences, Berlin, Germany

**Correspondence:** F. Miesner (frederieke.miesner@awi.de)

**Abstract.** The thermal regime in the sediment column below shallow water bodies in Arctic permafrost controls benthic habitats and permafrost stability. We present a robust, portable device that measures detailed temperature-depth-profiles of the near-surface sediments in less than 1 hour. Test campaigns in the Canadian Arctic and on Svalbard have demonstrated its utility in a range of environments during winter and summer. Measured temperatures were spatially heterogeneous, even within single
water bodies. We observed the broadest temperature range in water less than 1 m deep, indicating that the bottom-fast ice zone is overlooked by single measurements in deeper water.

## 1 Introduction

Lakes cover approximately $6\%$ of the land cover in the Arctic, with small lakes (area smaller than $0.01\,\mathrm{km}^2$) making up about $32\%$ of the lakes (Paltan et al., 2015). Water bodies in the permafrost region in the Arctic cause thermal disturbances
in comparison to the surrounding ground (i.e. Boike et al., 2015; Jorgenson et al., 2010; Burn, 2002). As the mean annual water temperature is usually above $0\,°\mathrm{C}$, even if the mean annual air temperature is below $0\,°\mathrm{C}$, unfrozen zones within the permafrost, called taliks, can form below the water bodies. However, the presence of bottom-fast ice in areas with water shallower than the maximum ice thickness can decrease the mean annual bed temperature and significantly slows thawing or even refreezes the lake or sea bed in winter (Roy-Leveillee and Burn, 2017; Arp et al., 2016). In the transition zone between
floating and bottom-fast ice, small changes in water level have the potential to drastically alter the sub-bed thermal regime between permafrost-thawing and permafrost-forming.

The temperature regime of lake sediments is a determining factor for microbial activity. Among other factors, warm temperatures can render taliks hot spots of methane gas emission Abnizova et al. (2012). A better understanding of the thermal regime with temperature measurements in a fine spatial mesh can therefore help to better constrain the emission potential of shallow
Arctic water bodies. However, such distributed measurements of sediment temperatures are scarce. *Temperature lances* have been designed for measuring in-situ temperatures, and even sediment thermal properties (Lister, 1970; Hyndman et al., 1979; Sclater et al., 1969; Christoffel and Calheam, 1969), but their use in the Arctic, and especially under small lakes, requires portability and robust design. Due to lake/sea ice dynamics and ice movement during the melt period, monitoring with permanently





installed devices is generally not feasible in the Arctic. Commonly used in offshore marine environments are heat flow probes of Bullard type (the sensor string is directly attached to the logger and pushed into the sediment, Christoffel and Calheam (1969); Lindqvist (1984)), Lister or violin-bow type (the sensor string is attached like a violin bow to a solid strength member that is lowered into the sediment, Lister (1970); Hyndman et al. (1979)), or Ewing type (outrigger fins with sensors are attached to a piston or gravity corer, Riedel et al. (2015)). These devices are heavy and require larger ships for deployment (Hornbach et al., 2021) and surveys are typically focused more on thermal properties and the geothermal heat flow than on the temperature field alone Dziadek et al. (2021). Furthermore, they are not useful to small lakes or the shallow waters of the near-shore zone due to their weight and unwieldy design. Smaller lances are being used to monitor sediment temperatures in permafrost areas (Dafflon et al., 2022), but are not suitable for submergence. Published temperature-depth profiles in the near-shore zone and under shallow lakes have been recorded with temperature chains in boreholes (i.e. Solomon et al., 2008; Brown and Johnston, 1964; Burn, 2002). Although this yields good quality data to potentially great depths, single-location temperature-depth profiles hold no information about the spatial variability of temperatures. In addition, although boreholes afford a chance to repeat measurements or even measure a continuous time-series, in aquatic environments they are often disturbed or destroyed by ice movement. Lindqvist (1984) used a Bullard-type probe to monitor lake bed temperatures in northern Sweden, but the device was also not portable enough to make spatially distributed measurements. An even smaller probe was developed and used to monitor temperatures in the upper 40 cm of the tidal plains in the German Wadden Sea (Onken et al., 2010), where deployment is easy during low tide.

We therefore suggest that a lightweight robust device to quickly measure temperature profiles under shallow lakes in the Arctic could fill a knowledge gap addressing spatial variability of sediment temperatures in shallow water.

Here we present a newly developed temperature lance and its applicability in a number of Arctic environments, providing a detailed technical description of its design, construction and intended modes of operation, as well as post-processing of the data, before showcasing collected data sets to demonstrate the device's utility.

## 2  Methods

### 2.1  Technical Description

The temperature lance is built from high-grade stainless steel, which ensures its durability and resistance to corrosion. The lance is 1.5 m long and 20 mm in diameter. The PCB string comprises 15 digital nodes with two temperature sensors each, resulting in 30 temperature sensors with a 5 cm spacing. The temperature nodes are connected end-to-end forming a common communications and power bus. The digital temperature sensor used is a `TSYS01-1` (G-NICO-023, TE Connectivity) with an accuracy of 0.1 °C and resolution of 0.01 °C.

To shorten equilibration time with the surrounding sediment, small copper cylinders couple the temperature sensor through the body of the lance to the outside environment 1. The thermal conductivity of the copper ($\kappa_{copper} \approx 360 \, \mathrm{W \, m^{-1} \, K^{-1}}$) is approximately 20 times higher than that of the stainless steel body ($\kappa_{steel} \approx 18 \, \mathrm{W \, m^{-1} \, K^{-1}}$), ensuring that the vertical heat conduction is negligible compared to the horizontal component.



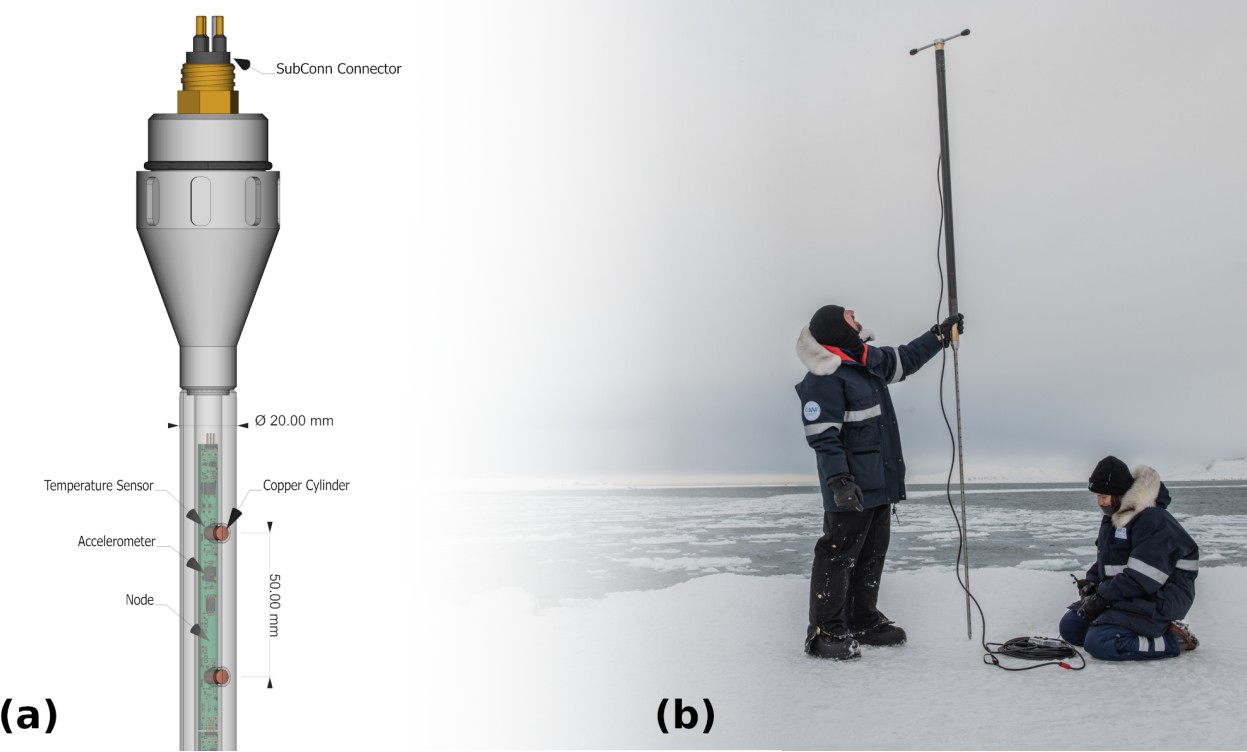

**Figure 1.** (a) Technical drawing of the head of the temperature lance. A string of 15 nodes is housed inside the lance, with copper cylinders protruding the stainless steel body, connecting the temperature sensors with their surroundings. Two temperature sensors are mounted on each node; on two nodes (at the top and in the middle of the lance) an additional accelorometer is mounted. (b) The temperature lance assembled with one carbon fiber extension for deployment in up to 1.5 m deep water, and connected with a water-proof cable to the logger unit.

Accelorometers are installed on two of the nodes at the top and in the middle of the lance to measure the tilt. This information can be used during deployment to assess departure from vertical, and can be used in post-processing to correct sensor depths.

Finally, the lance body is filled with a urethane-potting compound to protect the electronics.

Measurements are conducted with an Arduino-based logger containing a GNSS module to obtain accurate recordings of date, time and position for each measurement. Communication with the temperature nodes is realized via 1-Wire protocol (Maxim Integrated / Analog Devices), which allows for cable lengths of over 100 m. The logger records the temperature measurements onto a micro SD card every 30 – 240 s by default, but the interval is adjustable. During an active measurement, the logger unit indicates whether the most recent two measurements at each sensor have not changed more than a certain threshold. We

suggest a threshold of 0.02 K and a measurement interval of 30 s, but both can be adjusted. The GPS receiver is located in the logger unit and the position is written to the data file only once at the beginning of each measurement, i.e. the logger unit must be located at the measurement location when starting a measurement.





## 2.2 Deployment Options

The temperature lance is inserted into the sediment, usually by hand, and then left undisturbed for several minutes until the
temperatures of the sensors are in equilibrium with the surrounding sediment, i.e. until the maximum temperature-change
threshold condition is satisfied. The resistance to insertion and the equilibration time depend largely on the sediment composi-
tion. The device is robust enough to allow hammered insertion if needed. The options for deployment include through a hole
in ice cover, from vessels at the water surface (may require anchoring or stabilization), or from a wading position in shallow
water. Deployment is not restricted to aquatic environments, i.e. temperature measurements can be made in the active layer as
well as in taliks or any material loose enough to permit insertion. Lightweight hollow carbon fiber extensions are screwed onto
the temperature lance during the deployment in deeper water.

## 2.3 Post-processing of the Data

The calibration of each temperature sensor was checked at $0\,°\mathrm{C}$ in an ice-bath (Wise, 1988) and the offset from $0\,°\mathrm{C}$ was
measured before assembly of the lance. Measured temperatures are corrected by the logger unit at the time of measurement
using these offsets.

When the device enters the sediment it has a temperature close to air or water temperature, which may be several degrees
warmer or colder than the sediment, and frictional heat from the penetration of the measurement device heats up the sediment
(Carslaw and Jaeger, 1959). Both factors may perturb sediment temperature, meaning that measured sensor temperatures do
not reflect undisturbed in-situ sediment temperature. We employ a inversion scheme, following (Bullard, 1954; Lister, 1970;
Hartmann and Villinger, 2002), to determine in-situ temperatures. For each measurement, we leave the lance undisturbed in the
sediment until the threshold of $0.02\,\mathrm{K}$ between two consecutive measurements is not exceeded for at least four measurements.
The time series of temperature measurements is then fit to the theoretical function for temperature decay from a cylindrical
source of heat

$$\Theta(t) = \Theta_0 + \delta\Theta F(\alpha, \tau(t)) = \Theta_0 + \delta\Theta \int_0^\infty e^{-\tau(t)x^2} f(x)\,dx, \tag{1}$$

where $\Theta$ is the temperature over time $t$, $\Theta_0$ is the undisturbed sediment temperature, $\delta\Theta$ the disturbance introduced by the
temperature lance and $\tau = \frac{\kappa t}{a^2}$ (Bullard, 1954). The variable $\alpha = 2\pi a^2 \rho\sigma/m$ is a composite of the thermal properties of the
lance body ($m = 360\,\mathrm{W\,m^{-1}\,K}$ the thermal conductivity and $a = 0.0095\,\mathrm{m}$ the length of the copper pieces) and the sediment
(the effective heat capacity $\sigma$, and the density $\rho$, to be measured or estimated at each location). The function $f$ is integrated
over the radial coordinate $x$ and is written with the Bessel functions $J$ and $Y$ to be

$$f(x) = \frac{4\alpha}{\pi^2 x} \left[ \{xY_0(x) - \alpha Y_1(x)\}^2 + \{xJ_0(x) - \alpha J_1(x)\}^2 \right]. \tag{2}$$

We use a least-squares algorithm to fit the measured temperatures over time to Eqn. (1) with the desired in-situ temperature
$\Theta_0$, the introduced disturbance $\Theta_1$, as well as the thermal properties of the sediment as free variables.



## 2.4 Study Sites

We provide a comprehensive analysis of sediment temperature measurements obtained from various Arctic study sites: a) the
beach zone of *Tuktoyaktuk Island* in the harbour of Tuktoyaktuk, Northwest Territories (NWT), Canada, b) a shallow lagoon
near Ny-Ålesund, Svalbard, c) an old river arm in the outer Mackenzie Delta, d) a small thermokarst lake near the Inuvik-
Taktoyaktuk-Highway, NWT, Canada and e) the drinking water reservoir of Ny-Ålesund, Svalbard.

a) *Tuktoyaktuk Island* is a barrier island in the harbour of the hamlet of Tuktoyaktuk at the Arctic Coast in the Northwest
Territories, Canada. The coastline of *Tuktoyaktuk Island* has changed dramatically over the last decades due to coastal erosion,
especially the north-facing shore moves slowly southwards with 1 – 2 m/yr (Whalen et al., 2022).

b) and e): We accessed the lagoon, *Brandalaguna*, and the drinking water lake, *Tvillingsvatnet*, near Ny-Ålesund by snow-
mobile in March 2021. The lagoon has a shallow water depth of 1.4 m and brackish water with salinity around 2 – 3 psu. The
drinking water lake has a maximum water depth greater than the 4.6 m where we measured. We carried out the equilibration
experiment at the drinking water lake. Both water bodies were covered by ≈ 60 cm thick ice.

c) The old river arm in the outer Mackenzie Delta is sometimes called *Swiss-Cheese Lake*, because ebullition, even in winter,
maintains holes in the ice cover. The lake is reachable by boat and a 2 km hike or by helicopter in summer, or by snowmobile
from the ice-road in winter.

d) The small thermokarst lake, dubbed *Lake 3* throughout this paper, is located about 300 m from the Inuvik-Tuktoyaktuk-
Highway and therefore easily accessible throughout the year.

## 3 Results

The device was tested on multiple expeditions in the Arctic in summer as well as in winter. Operation through a hole in the ice
and from a wading position in late summer proved to be executable by a single person. Assistance was useful to apply more
weight when pushing the device into the sediment but not strictly necessary. Operating the device from small boats was always
a two-person job and was most successful in calm weather without wind and waves.

The visited sites cover freshwater and marine-influenced water bodies, with sediment grain sizes ranging from fine silty
sands to coarse gravel, and shallow and swampy as well as deeper water. In locations with coarser sand and gravel, the sediment
insertion depth was sometimes limited to 1 m.

The measured sediment temperature depth profiles range from nearly isothermal states to temperature gradients of more than
9 K m$^{-1}$. An overview of the site and measurement characteristics is given in Tab. 1.

## 3.1 Post-processing of the Data

To assess the accuracy of the results of the inversion algorithm, we conducted a 12 h measurement in March 2021 at the
drinking water lake near Ny-Ålesund, Svalbard. The measurement was taken in 4 m water depth under floating-ice conditions
We assume that there was no diurnal variability in the bottom water temperature at that water depth, so that the mean of



the measurements after 12 h reflected the in-situ temperature. With this measurement we compared the inversely-determined
in-situ temperatures with the mean over three consecutive measurements and explored how the accuracy changes with time
following beginning of the measurement. Fig. 2 shows the maximum absolute error over the whole temperature depth profile
for the mean of three measurements (blue crosses) and the inversely determined results (pink circles). The error of inversely
determined temperature-depth profiles was smaller by $\approx 0.01\,\mathrm{K}$ than for the means for short measurement times shorter than
45 min. Both yielded results within the sensor accuracy of $0.01\,\mathrm{K}$ after $\approx 80\,\mathrm{min}$. For even longer measurement time, the means
are more accurate than the inversely determined values.

The time needed to equilibrate to in-situ temperatures depends on the thermal properties of the sediment as well as the
initial temperature difference between the temperature lance and the sediment. This assessment of error over time from this
one long-term measurement is therefore only qualitative. Using the inversion algorithm improves the accuracy of the results in
comparison to taking the mean, at least at shorter measurement periods. The time needed to achieve sensor accuracy, however,
cannot be determined universally from this experiment. We recommend recording temperatures while allowing the device to
equilibrate with the sediment for at least 30 min.

### 3.2 Spatial Heterogeneity

All temperature-depth profiles for the four locations are shown in Fig. 3.

a) The measurements in Tuktoyaktuk harbour were taken both south and north of the barrier island. They all show a similar
shape, with a temperature increase from the water temperature down to $\approx 50\,\mathrm{cm}$ into the sediment and then a steady decrease
below. All profiles suggest a probable upper permafrost boundary in $\leq 3\,\mathrm{m}$ sediment depth. In general, at this site, the deeper the
water, the higher the water temperature and also the sediment temperature, and the deeper the assumed depth of the permafrost
table.

b) The temperature profiles at *Brandalaguna* are the only ones that increased steadily over depth. They are also the only ones
measured in late March, when the air temperature was near its annual minimum. The shape of the temperature-depth profiles are
relatively similar to each other and the measurements are not deep enough to show a potential change in the gradient that would
indicate lower temperatures in deeper sediment layers. The water depth shows only little spatial variety, changing only within
20 cm. However, we still observe a temperature difference of $> 1\,\mathrm{K}$ at the same depths below the sediment-water-interface.

c) At *Swiss-Cheese Lake* the water temperature is relatively stable at $8.5 - 9\,°\mathrm{C}$. The measurements in water depth $> 0.9\,\mathrm{m}$
show a similar shape and indicate a deep talik. The two shallower measurements, located within 2 m of the lake shore, reach
frozen ground at around 1.2 m sediment depth.

d) At *Lake 3* we see two clusters of temperature-depth-profiles: First, measurements in water depths $> 0.9\,\mathrm{m}$ with water tem-
peratures of $\approx 6.3\,°\mathrm{C}$. Here, the sediment temperature increases up to $\approx 60\,\mathrm{cm}$ sediment depth and decreases below, pointing
to a talik depth of $\geq 5\,\mathrm{m}$. And second, measurements in shallower water depth with water temperatures between 1.2 and $4.5\,°\mathrm{C}$,
where the temperature-depth profiles show great variability. The four measurements in water depth below 0.3 m reach frozen
ground within the upper metre of sediment, while the other measurements point to a deeper talik than even the measurements
in the deeper parts of the lake.



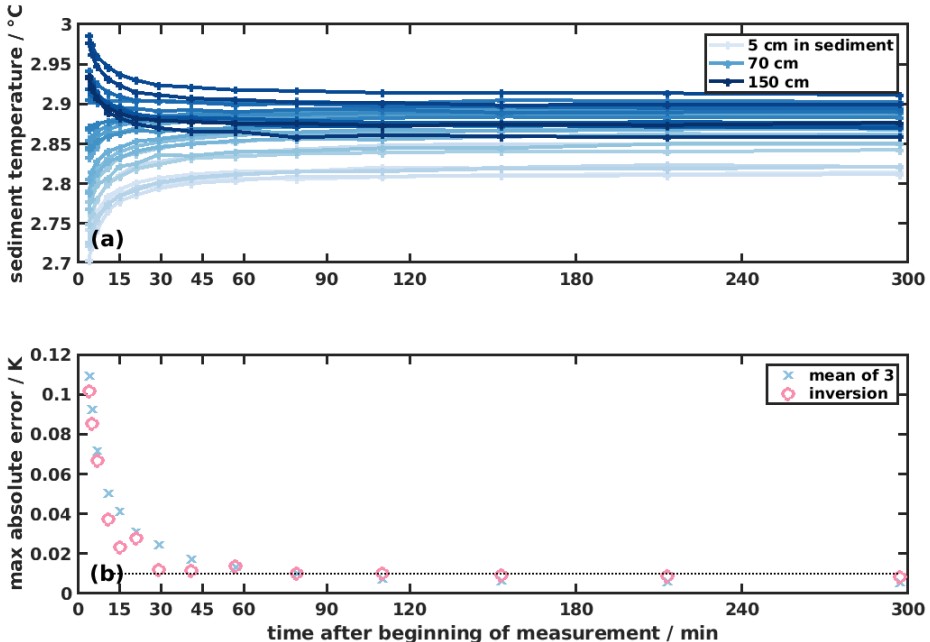

**Figure 2.** (a) Temperature over time after the beginning of the measurement at *Tvillingsvatnet*, Svalbard, for all sensors. The shade of blue indicates the depth of the sensor in the sediment, with lighter shades at the water-sediment interface and darker shades towards 1.5 m deep in the sediment. Measurements were taken every 60 s. The 14 highlighted points in time (+) indicate the timing of the time periods for which we employed the inversion algorithm. (b) Maximum temperature difference over all sensors for results at different times after the beginning of the measurement. The mean over three measurements after 12 h is assumed as the in-situ temperature, i.e. the reference temperature that the results are compared to. The results from the inversion algorithm are shown in pink circles, means over three measurements are shown in blue crosses, the dotted line indicates the sensor accuracy of 0.01 K.

We observe, that measurements at all locations reveal spatial heterogeneity in sediment temperatures, as well as difference between the locations even with the same water depth. Water depth alone is not able to cluster the measured temperature-depth profiles over all sites.

## 4  Conclusions

Our newly built temperature lance has proven suitable for use in a variety of Arctic aquatic environments. It is portable enough to be brought along on helicopter flights and carried by foot over the tundra. It was successfully used in winter from lake ice as well as in summer from a rubber boat and wading in shallow water. The maximum penetration depth when hand-operated varies with the structure of the sediment, with fine sands being ideal. Hammering the lance was only moderately successful in increasing penetration depth. Extracting the lance was easiest from the boat, using the steady up-and-down rhythm of the





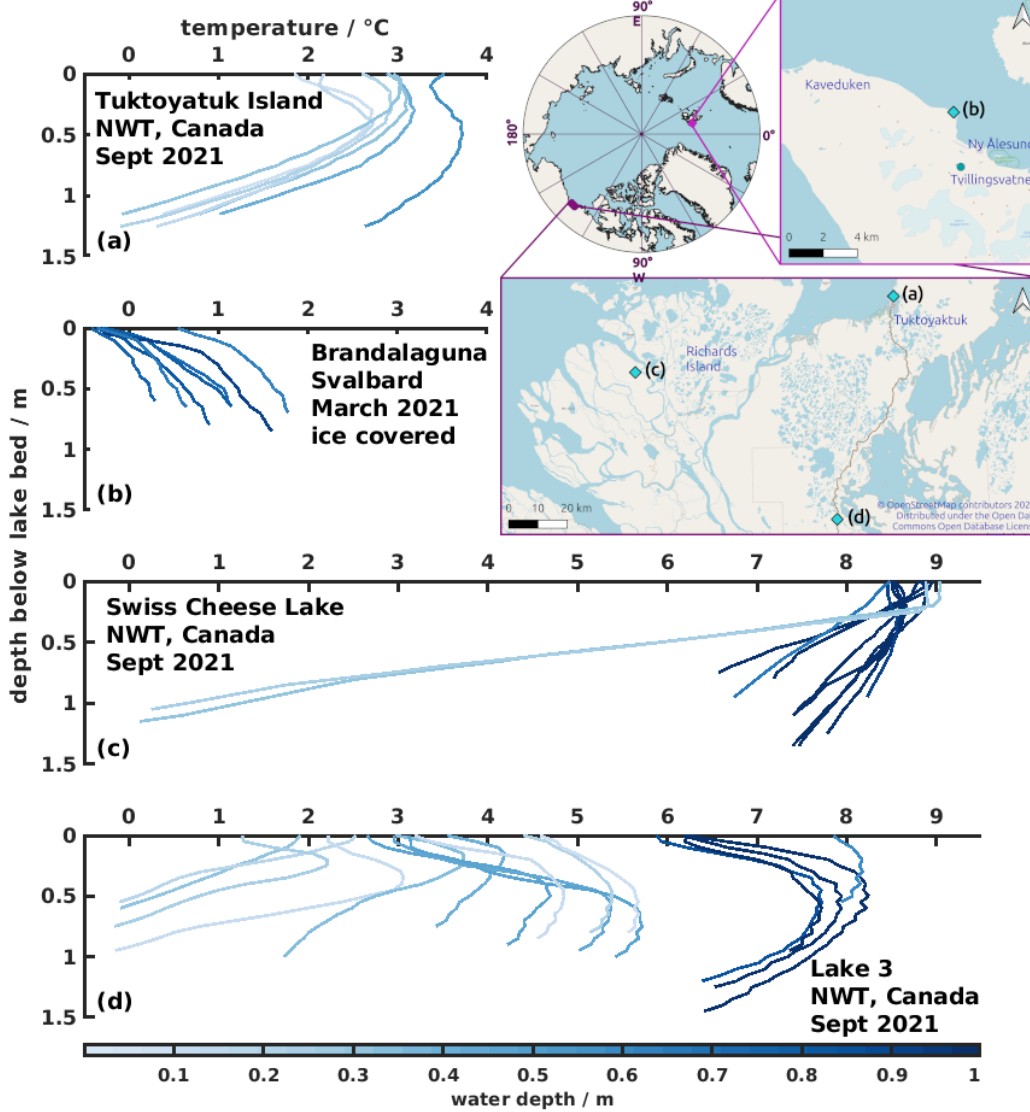

**Figure 3.** Temperature depth profiles grouped by investigation sites. The map shows all locations in teal diamonds and additionally the drinking water lake of the long-term measurement as a circle. Map data is from © OpenStreetMap. (a) Measurements below the Arctic Ocean at *Tuktoyaktuk Island*, NWT, Canada. Measurements were taken from a wading position in the water in September 2021. (b) Measurements at *Brandalaguna*, Svalbard, were taken through ≈ 80 cm ice cover in March 2021. (c) Measurements at *Swiss-Cheese Lake* in the outer Mackenzie River Delta, NWT, Canada, were taken from a small rubber boat in September 2021. (d) Measurements at *Lake 3*, near the Inuvik-Tuktoyaktuk-Highway, NWT, Canada, were taken from a small rubber boat in September 2021. *Brandalaguna* and *Tuktoyaktuk Island* have brackish water, while the other two are fresh water lakes. The shades of blue indicate water depth and the scale is the same over all sites. We measured in water as deep as 2.5 m (at *Swiss-Cheese Lake*), but the color scale is cropped to 1 m to better distinguish between shallower values.





boat in the waves. This indicates that vibrating the lance in, either with a motor or a drop weight, could be another option for deployment with more reliable deeper penetration.

The equilibration time depended largely on the sediment's thermal properties and the initial temperature gradient between the lance and the sediment, but was overall short enough to be able to acquire measurements in under one hour, including the time needed to push the lance into the sediment and to recover it. We recommend leaving the device in the sediment for more than 30 min in the sediment to ensure sufficient thermal equilibration with the sediment. Using an inversion algorithm to retrieve an estimate of the asymptotic in-situ temperatures improves the accuracy of the resulting temperature-depth profile, especially for short measurements, but this advantage fades when the measurement is long enough to have good accuracy.

The presented data shows significant variation between the spatially distributed temperature-depth profiles obtained at each site and captured spatial heterogeneity that would be unavailable from single boreholes. We observe similar temperature-depth profiles in the deeper water ($> 1\,\mathrm{m}$) at each location, and large variability (up to 5 K) in shallower water ($< 0.9\,\mathrm{m}$).

The importance of the spatial heterogeneity in shallow water is possibly linked to zones with bed-fast ice. Permafrost may be found under lakes with bed-fast ice regimes (Arp et al., 2016), while floating ice lakes may develop through taliks (Boike
et al., 2015; Arp et al., 2016; Burn, 2002). Burn (2002) monitored lake water and sediment temperatures during 1992 – 1996 at various lakes on Richards Island, Northwest Territories (see Fig. 3). They found permafrost below the terraces with water depth shallower than $\approx 60\,\%$ of the maximum ice thickness, i.e. shallower than 1 m. The measured mean annual lake-bottom water temperatures in these shallow parts varied, however, over a wide range from $-6$ to $0\,°\mathrm{C}$. Our observations at *Swiss-Cheese Lake* corroborate these findings Lake sediment temperatures in shallow waters tend to higher spatial heterogeneity. Single-
location sediment temperature profiles, for example, from boreholes, cannot capture the variety in the temperature field. Our measurements can capture this variety, but would require repeat deployments to capture temporal variability.

The resulting data sets will bring new insights into the temperature dynamics under shallow lakes, especially in the bottom-fast ice zone. Data from locations like *Swiss-Cheese Lake* can help to better constrain patterns of methane emissions and support investigations of the sources (Wesley et al., 2022). These data are suitable as validation for modeling, can give additional spatial
information around borehole installations and may yield insights into heat fluxes at the sediment-water interface. Future work will focus on estimating permafrost table from these measurements via extrapolation and thermal modeling.

*Data availability.* The data is available from Pangaea, the references are provided in Tab. 1.

*Author contributions.* FM, WLC, PPO and JB conceptualized the design of the lance and the fieldwork. WLC constructed the devices with the assistance of FM and wrote the logger code. FM, WLC and JB carried out the field work. FM analyzed the data and wrote the manuscript.
JB and PPO acquired the funding. All authors contributed to the final manuscript.





*Competing interests.* The authors state no competing interests.

*Acknowledgements.* This work is part of the Helmholtz Association in the framework of MOSES (Modular Observation Solutions for Earth Systems). The authors would like to thank the Scientific Workshop of the Alfred Wegener Institute in Bremerhaven for design assistance and manufacturing the lance body and extensions.



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




**Table 1.** Summary of the covered sites of investigation. The water parameters were measured with a CastAway CTD prior to each measurement with the temperature lance. The data sets are individually available via PANGAEA.

| site name | Tuktoyaktuk Island | Brandalaguna | Swiss-Cheese Lake | Lake 3 | Tvillingsvatnet |
|---|---|---|---|---|---|
| area | Arctic Coast, NWT, Canada | Svalbard | Outer Mackenzie Delta, NWT, Canada | NWT, Canada | Svalbard |
| lat | 69.455 | 78.9448 | 69.233 | 68.7756 | 78.9161 |
| lon | -133.0033 | 11.8583 | -135.2551 | -133.5410 | 11.8757 |
| date | 28 Sept 2021 | 25 March 2021 | 13 – 16 Sept 2021 | 18 – 23 Sept 2021 | 26 March 2021 |
| max water depth / m | 0.9 | 1.45 | 2.9 | 2.2 | 4.6 |
| mean water depth / m | 2.8 | 0.1 | 8.8 | 4.8 | 2.8 |
| mean salinity / psu | 2.5 | 3 | 0.1 | 0.05 | 0.2 |
| data set | (Miesner et al., 2022f) | (Miesner et al., 2022a) | (Miesner et al., 2022e) | (Miesner et al., 2022d) | (Miesner et al., 2022a) |
| sediment temp. range °C | -0.2 – 3.7 | -0.4 – 1.7 | -0.1 – 9.1 | -0.2 – 8.1 | 2.7 – 2.9 |
| max sensor depth range m | 1.10 – 1.30 | 0.75 – 0.95 | 0.50 – 1.40 | 0.85 – 1.35 | 1.45 |
| data set | (Miesner et al., 2022c) | (Miesner et al., 2022b) | (Miesner et al., 2022c) | (Miesner et al., 2022c) | (Miesner et al., 2022b) |