# Peer review of "Brief communication: Testing a portable Bullard-type temperature lance confirms highly spatially heterogeneous sediment temperatures under shallow water bodies in the Arctic"

_The Cryosphere, 2023_

## Author Response (AR1)

**Editors Comment**

Dear authors,

we have received the evaluation of the external referees and their comments are very positive about the manuscript. The suggestions are rather on the formulations and phrasing without corrections from technical aspects. The inclusion of the suggestions seems to be doable from the reply provided. Hence, my decision is to accept the paper with minor revisions (without further evaluations by external reviewers).

From the answers given, my only suggestion is also to consider the comment of the reviewer 2 and provide maybe a few lines on the way the accelerometer is built-in and the way it works. Besides that, please implement your revisions as proposed in your reply to the reviewers.

I am looking forward to read your revised version of the manuscript.

Best regards,

Adrián Flores Orozco

**Thank you very much for the evaluation of our manuscript. We have worked the Reviewers comments into the manuscript as indicated in our final responses. Please find below the updated answers to the reviewer, indication where and what exactly we changed.**

**Reviewer 1**

I carefully read the Manuscript entitled "Brief communication: Testing a portable Bullard-type temperature lance confirms highly spatially heterogeneous sediment temperatures under shallow water bodies in the Arctic" by Miesner et al. Good manuscript, good technical paper.

**Thank you very much for your review. We appreciate your comments and will do our best to incorporate them into the revised manuscript. Especially the notes on discussing the equations in the post-processing will improve the understandability of the manuscript. We will also add a sentence explaining the logger unit and the connection in more detail.**

I have a couple of minor edits :

Line 17 : I would slighlty increase the complexity of the sentence, by adding a reference or an example on microbial activity in lakes. Otherwise I would remove it, since you do it good for methane emissions.

**Agree, we dropped the sentence with microbial activity, as this**

**is not the focus of this paper but just the motivation for the measurements.**

Line 18 : "methane gas emission". I would remove gas, since methane is a gas in these external condition

**Agree.**

Line 18-20: Are we looking at the thermal regime here, or finally the vertical temperature spatial variability ? Unsure we are looking at time series to quantify the regime here.

**Thanks for pointing out this ambiguity. We are talking about the thermal regime here, in what would be desirable (knowledge of the spatial and temporal variability of sub-lake sediment temperatures). The temperature lance can be easily used to measure the spatial variability and this is what we show in the paper. With repeat measurements at different times, one could also use the device to cover temporal variability but that is indeed not part of this paper. The sentence now reads** *A better understanding of the spatial variability with temperature measurements in a fine mesh can therefore help better constrain the emission potential of shallow Arctic water bodies.*

Line 26 : "Lister (1970)'s ?"

**Yes and no. This type of heat flow probe is called Lister type (or violin-bow type) after Lister (1970), which we cite in line 27.**

Line 30 : Should be "(Dziadek et al. 2021)"

**Yes.**

Line 51-52 : You use the 24 bit resolution ?

**Yes.**

Line 54 : " the outside environment 1". You meant Figure 1 I suppose, otherwise your fig is not called.

**Yes, thank you for catching this.**

Line 59 : " urethane-potting compound to protect the electronics." I would add that this electrical insulating linner has a lower heat conductivity than all the rest of the assembly. Even if it seems obvious.

**We added a half-sentence in the revision (line 60 of the revised manuscript).**

Line 65 :" 0.02 K Threshold", might be good to keep the Celsius unit here since you are mentionning all your variable in C ?

**We agree that it is less confusing to stick with degree C for all. We will adopt this is the review.**

Part 2.1 - 2.2 :

- If you could add a small sentence explaining how you communicate with the probe (No need to be extremely exhaustive). Is it based on Ethernet

connection (?), bluetooth, wifi ? Or you simply set them up prior to field deployment ? If so, how/what is the communication protocol ? IN Figure 1-b) we see the logger unit, is it a computer ? - If you could add a small description about the power supply/battery. On figure 1-b), I think we see an external power supply ?

**Thanks for pointing out that this is not adequately explained. We will add a clarifying sentence in part 2.1, starting line 64 of the revised manuscript.**

-Line 90-93. Could you try to define only one side of the equation. It is difficult to follow, since sometime you are describing the RHS of the equation with "f" and in some other part you are describing alpha. Maybe only focus on the RHS directly stating that f(x), is actually f(x,alpha), and then your paragraph is ok to follow.

**We agree that this paragraph is rather hard to follow. We will rewrite this to make it more clear what is happening in the review. The new paragraph reads as follows (starting line 91 of the reviewed manuscript):** *The time series of temperature measurements is then fit to the theoretical function for temperature decay from a cylindrical source of heat*

$$\Theta(t) = \Theta_0 + \delta\Theta \int_0^\infty e^{-\tau(t)x^2} f(\alpha, x)\, dx, \tag{1}$$

*where $\Theta$ is the temperature over time $t$, $\Theta_0$ is the undisturbed sediment temperature, and $\delta\Theta$ the disturbance introduced by the temperature lance with the integral describing the decreasing amplitude of this disturbance with distance to the lance body (Bullard, 1954). The variable $\alpha = 2\pi a^2 \rho\sigma/m$ is a composite of the thermal properties of the lance body ($m = 360\,\mathrm{W\,m^{-1}\,K}$ the thermal conductivity and $a = 0.0095\,\mathrm{m}$ the length of the copper pieces) and the sediment (the effective heat capacity $\sigma$, and the density $\rho$, to be measured or estimated at each location), and $\tau = \frac{\kappa t}{a^2}$ is a scaled time variable. The function $f$ is integrated over the radial coordinate $x$ (i.e. the distance to the lance body) and is written with the Bessel functions $J$ and $Y$ to be*

$$f(\alpha, x) = \frac{4\alpha}{\pi^2 x} \left[ \{xY_0(x) - \alpha Y_1(x)\}^2 + \{xJ_0(x) - \alpha J_1(x)\}^2 \right]. \tag{2}$$

*We use a least-squares algorithm to fit the measured temperatures over time to Eqn. (1) with the in-situ temperature $\Theta_0$, the introduced disturbance $\Theta_1$, as well as the thermal properties of the sediment as free variables.*

Line 134 : Sensor accuracy of 0.01 K. Again, unsure this is good to mix Kelvin and Celsius.

**We have adopted K throughout the manuscript.**

Figure 2 : I would re-draw the legend, to clearly see that the shade of blue mimics depth. A simple colorbar should do the job ? As you did for fig 3 ?

**Thanks for this suggestion, we adopted this.**

Line 183-187, might need to be moved to the part 3, since you are really discussing your data in regards with the literature.

**Thank you for this suggestion, we moved part of this discussion up, now starting now at line 176 of the reviewed manuscript.**

**Reviewer 2**

This study presents the development of a portable Bullard-type temperature lance, and its application to capture spatially heterogeneous sediment temperatures under shallow water bodies in the Arctic. I enjoyed reading the manuscript. I think the work is relevant and of interest for the community, and adequate to be presented as a brief communication article. I only have some minor comments here below.

**Thank you very much for your review and your valuable comments. We will carefully rephrase paragraphs and add information where you indicated we were a bit too short on details. We will add a paragraph discussing how we estimate the penetration depth of the lance. We will also add more detail to the description of the technical details of the PCBs.**

L.6: The meaning of "overlooked" is not clear (to me at least). Do the authors mean that a single shallow measurement of water temperature would have failed in detecting the presence of bottom-fast ice zone. Please consider rephrasing for improving clarity.

**That is exactly what we mean, the sentence now reads** *We observed the broadest temperature range in water less than 1 m deep, a zone that is not captured by single measurements in deeper water.*

L.8: Consider rephrasing with "The portion of land covered with lakes is ..."

**Adopted.**

L.13 and 14: if "slows" and "refreezes" refer to "can", remove the "s" at the end.

**They refer to the presence of bottom-fast ice. The sentence now reads (starting line 12 of the reviewed manuscript)** *However, in areas with water shallower than the maximum ice thickness the presence of bottom-fast ice can decrease the mean annual bed temperature and significantly slows*

*thawing or even refreezes the lake or sea bed in winter.*

L.18, L.30 and likely other places: The reference should be in parenthesis.

**Thanks for catching this. Adopted.**

L.16: the "to" before "better" can be removed.

**Assuming this is line 19, yes, adopted.**

L.49: Please consider providing more information about the PCB boards and the connectors, including if the sensors are mounted on a single long board or if boards are connected together (and if yes, how). Also consider providing information on how the SubConn connector is linked to the PCB.

**All connections between nodes and between the uppermost node and the SubConn connector are soldered (line 49 of reviewed manuscript).**

L.58: please consider describing how the accelerometers work, and adding later in the text, any findings that may be of interest for the reader. With regard to discussion of accelerometer and PCB connectors, here is a study that may be of interest: Wielandt, S.; Uhlemann, S.; Fiolleau, S.; Dafflon, B. Low-Power, Flexible Sensor Arrays with Solderless Board-to-Board Connectors for Monitoring Soil Deformation and Temperature. Sensors 2022, 22, 2814. https://doi.org/10.3390/s22072814

**Thank you for the literature suggestion. In the current iteration of the device, we really only used the accelerometer to see if the lance is upright when pushing it into the sediment. We added the following text (starting line 56 in the reviewed manuscript):** *Digital 3-axis accelerometers (ADXL345, Analog Devices) are installed on two of the nodes at the top and in the middle of the lance. The accelerometers measure the static acceleration of gravity from which inclination can be calculated. The high resolution of the sensor allows measurement of inclination changes of less than* $1.0°$*.*

L.96: "desired" sounds weird to me. If I understand correctly, the equation 2 is solved to estimate the in-situ temperature. Please consider clarifying which parameter is estimated vs assumed to be known. And consider adding a sentence in the results to explain their impact on the estimated values shown on the figure 2b.

**Desired does sound a bit weird. We will rephrase this and make it more clear in the review.**

L.108: "water depth greater than the 4.6m where we measured". Was the bottom of the probe in water and not in sediment? it does not look like to be the case based on Figure 3. Please improve clarity (sorry if I missed something).

**That sentence is indeed badly phrased. The lance was completely in the sediment. At that location the water was 4.6m deep. The maximum lake depth however, is greater than that. The sentence now reads (starting line 116 in the reviewed manuscript):** *At the location where we measured in the drinking water lake, the water was* $4.6\,\mathrm{m}$ *deep; however, the maximum water depth is greater than that.*

L.115: Consider adding some discussion about the uncertainty in estimating the lake depth. In Figure 3, is the 0 on the y-axis defined based on a change in the temperature profile ? or is that depth defined based on resistance when pushing the probe when entering the lake bed ?

**This is a great comment and we will add some text on this. The 0 on the y-Axis is based on a mixture of change in the temperature profile in the final data and an estimate made in the field with independent water depth measurements (with a CTD). We added a paragraph starting line 95 of the reviewed manuscript:** *As the lance has no pressure sensor, the penetration depth has to be estimated based on additional observations in the field and in the time-series of the measurements. We measured the water depth with an independent device and with the lance and the used extensions, to get the penetration depth in the field. The difference between water and sediment is also visible in the recorded data. With the* $5\,\mathrm{cm}$ *spacing of the temperature sensors, this yields an error of the same amount.*

L.153 and L.182: consider using Celsius only.

**We adopted degree Celsius in the whole manuscript.**

L.161: upper meter

**Yes, thank you, added.**

L.161: "than even the measurements" is not clear to me. Do the authors mean the deepest measurement in the lake ? please improve accordingly.

**Yes,the sentences now read (starting line 170 of the manuscript):** *The four measurements in water depth below* $0.3\,\mathrm{m}$ *reach frozen ground within the upper meter of sediment, while the other measurements point to a talik deeper than the measurements in the deeper parts of the lake would suggest. The temperature-depth profiles are therefore not deep enough, that a simple extrapolation could be used to estimate the depth of the talik at this location.*

L.189: consider adding "show" before "a higher spatial heterogeneity".

**Adopted.**